# A supramolecular system that strictly follows the binding mechanism of conformational selection

Liu-Pan Yang [1], Li Zhang[1], Mao Quan[1], Jas S. Ward [2], Yan-Long Ma[1], Hang Zhou[1], Kari Rissanen [2] & Wei Jiang [1✉]

Induced fit and conformational selection are two dominant binding mechanisms in biology. Although induced fit has been widely accepted by supramolecular chemists, conformational selection is rarely studied with synthetic systems. In the present research, we report a macrocyclic host whose binding mechanism is unambiguously assigned to conformational selection. The kinetic and thermodynamic aspects of this system are studied in great detail. It reveals that the kinetic equation commonly used for conformational selection is strictly followed here. In addition, two mathematical models are developed to determine the association constants of the same guest to the two host conformations. A "conformational selectivity factor" is defined to quantify the fidelity of conformational selection. Many details about the kinetic and thermodynamic aspects of conformational selection are revealed by this synthetic system. The conclusion and the mathematical models reported here should be helpful in understanding complex molecular recognition in both biological and synthetic systems.

[1] Shenzhen Grubbs Institute, Department of Chemistry, Guangdong Provincial Key Laboratory of Catalysis and Academy for Advanced Interdisciplinary Studies, Southern University of Science and Technology, Xueyuan Blvd 1088, Shenzhen 518055, China. [2] Department of Chemistry, University of Jyvaskyla, P.O. Box 35FI-40014 Jyväskylä, Finland. ✉email: jiangw@sustech.edu.cn

Molecular recognition[1,2] is ubiquitous in nature and is responsible for all biological processes. Understanding the fundamental mechanism of molecular recognition is central to understanding biology at the molecular level and is crucial for structure-based drug design, enzymatic catalysis and allosteric regulation of cell signaling[3,4]. In textbooks, induced fit, which was proposed by Koshland in 1958[5], is the dominant concept that has often been invoked to explain the conformational changes in molecular recognition. However, Monod, Wyman, and Changeux[6] proposed an alternative model—conformational selection—in 1965[7]. This model was largely overlooked at the time but has recently gained more experimental support in biology. For example, conformational selection has now been widely accepted to explain allostery[8] and signal transduction[9] in nature; the research on protein folding indicates that proteins exist as conformational ensembles[10], and thus, conformational selection may be a dominant mechanism in ligand binding of certain proteins[11]. The differentiation between these two mechanisms is often trivial[12] but important for understanding biological processes[4]. For some biological systems, the two mechanisms may coexist or gradually change from one to the other with different timescales[13].

These two limiting mechanisms assume different kinetic pathways (Fig. 1): in the induced-fit model, the ligand first binds to the receptor in a non-ideal conformation (L@H) and then induces the receptor to transition to the ideal conformation (L@H*). That is, binding precedes conformational changes; the ideal conformation may not be readily accessible in the absence of the ligand. In the conformational selection model, there are several discrete conformations (H and H*) of a receptor in equilibrium, from which the ligand selects the best fit (H*). In this case, a conformational equilibrium exists before binding. Consequently, kinetic experiments are the most compelling method to distinguish these two mechanisms. The rapid equilibrium approximation is often assumed[14], that is, the substrate exchange rate is considered to be much faster than the conformational interconversion rate. This is quite common for large and complex biological systems[15]. Under these conditions and according to the equilibria shown in Fig. 1, the observed first-order rate constants ($k_{obs}$) for the two mechanisms are different:[14]

Conformational selection:

$$k_{obs} = k_r + \frac{k_{-r}}{1 + \frac{k_{on}}{k_{off}}[L]} \tag{1}$$

Induced fit:

$$k_{obs} = k_r \frac{[L]}{\frac{k_{off}}{k_{on}} + [L]} + k_{-r} \tag{2}$$

$k_r$ and $k_{-r}$ are the forward and backward rate constants of conformational interconversion, $k_{on}$ and $k_{off}$ are the association and dissociation rate constants of ligand binding, and [L] is the ligand concentration. For conformational selection, $k_{obs}$ would decrease with [L], while $k_{obs}$ would increase with [L] and show saturation kinetics for the induced fit model. However, recent studies[16,17] indicate that an increase in $k_{obs}$ with [L] is not unequivocal evidence of the induced fit model[14]. This further complicates the assignments of the binding mechanism. In reality, the active conformation often exists at a low concentration, which is not even detectable; the inactive conformation can also bind the ligand. An ideal model for conformational selection should meet the following criteria: (a) two conformations coexist, and both can be detected; (b) only one conformation binds to the ligand, and the other one has no obvious affinity; (c) conformation exchange is kinetically slower than ligand binding. However, such

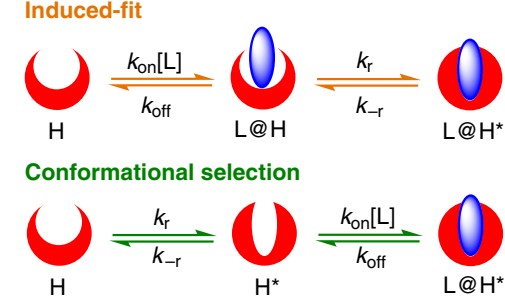

**Fig. 1 Two binding mechanisms involving conformational changes.** Induced-fit: ligand binding occurs before conformational change; conformational selection: conformational changes occurs prior to ligand bindng.

a clear-cut case with a conformational selection mechanism has never been reported in biological systems.

In contrast, detailed mechanisms of conformational changes in molecular recognition have rarely been studied in synthetic supramolecular systems[18–20]. Macrocyclic hosts are structurally simple and often have rather fast conformation exchange kinetics[21]. In addition, guest binding is also very fast[22]. The fast kinetics usually exceeds the detection limit of common kinetic techniques. Therefore, assignment of the binding mechanism is often inconclusive[18]. Induced fit is often directly assumed as the recognition mechanism of synthetic receptors[23]. When supramolecular systems become larger and larger, a conformational selection mechanism may become more important for understanding the binding behaviors. Very recently, Toste, Raymond, Bergman, and coworkers[19] reported the first and the only supramolecular system that follows the binding mechanism of conformational selection. The binding rate constant $k_{obs}$ of a self-assembled cage to tetraethylammonium was observed to decrease with increasing [L]. However, no further insight into conformational selection was obtained for this system.

In the present research, we report a macrocyclic receptor that strictly follows the binding mechanism of conformational selection. Macrocycle **1** (Fig. 2a) has two preexisting conformations that interconvert quite slowly; the binding mechanism is revealed to be conformational selection by simple NMR experiments and stopped-flow kinetic experiments. The mathematical equation (i.e., Eq. (1)) for the kinetics of conformational selection is strictly followed. In addition, two mathematical models are developed to simultaneously determine the association constants of the same guest to different conformations. A conformational selectivity factor, which is defined as the ratio between the association constants of the same guest to two conformations, is proposed to quantify the thermodynamic fidelity of conformational selection.

## Results

**Synthesis and characterization of macrocycle** 1. Bioreceptors not only have multiple conformations but also possess very high binding selectivity to certain guests. The conformational plasticity enables the cavity size and the cavity groups to adapt. The different selectivity of different conformations should depend on the difference in the arrangement of functional groups in the binding pockets. To realize conformational selection, the receptor should have multiple conformations that slowly exchange and can be easily distinguished by common techniques, such as $^1$H NMR spectroscopy. Additionally, the cavity should be decorated with polar functional groups, such as hydrogen bonding sites, to enhance the binding selectivity between different host conformations.

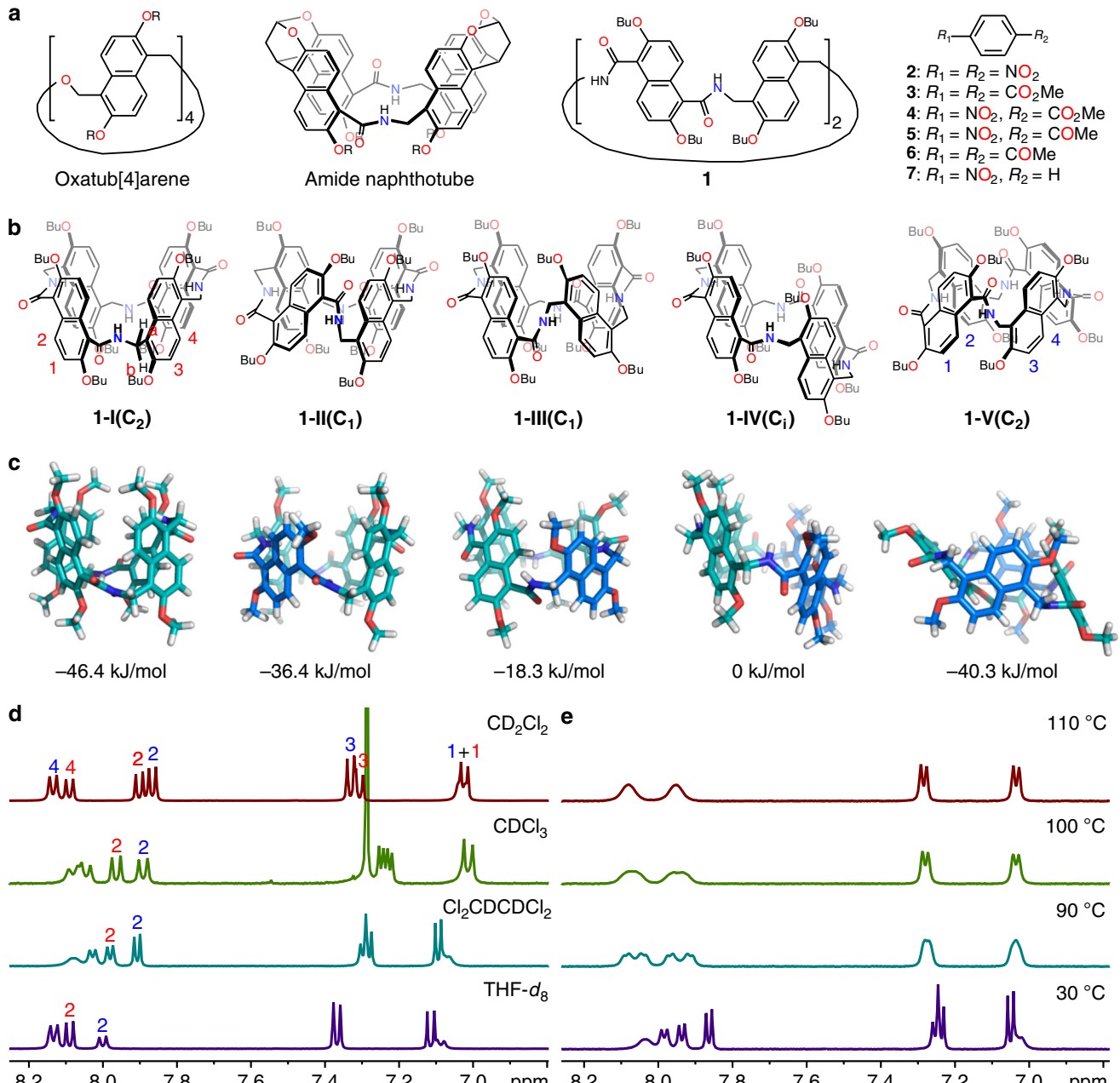

**Fig. 2 Chemical structures of all the related compounds and conformational analysis of macrocycle 1. a** Chemical structures of oxatub[4]arene, amide naphthotube, macrocycle **1** and all the guests involved in this research. **b** Chemical structures of the five conformers of macrocycle **1**. Colored numbering on the structures corresponds to the assignment of NMR signals for different conformations. **c** Energy-minimized structures of the five conformers calculated by DFT at the wB97XD/6-31 G(d) level of theory. **d** Partial [1]H NMR spectra (500 MHz, 25 °C) of **1** in different deuterated solvents. **e** Variable-temperature [1]H NMR spectra of **1** in Cl$_2$CDCDCl$_2$ (600 MHz).

Recently, we reported two macrocyclic hosts: oxatub[4]arene[24–27] and amide naphthotube (Fig. 2a)[28–32]. Oxatub[4]arene possesses four distinguishable conformations through naphthalene flipping, and different guest molecules select different conformations. However, the conformational change is too fast and does not satisfy the requirement for conformational selection. In addition, amide naphthotube has hydrogen bonding sites located inside its deep hydrophobic cavity and shows rather high binding selectivity, even in water. By combining the features of these two macrocycles (Fig. 2a), we may obtain a macrocycle with both slow conformational exchange kinetics and high binding selectivity. Accordingly, macrocycle **1** was designed and synthesized in this research (see Supplementary Methods).

Unlike oxatub[4]arene, macrocycle **1** has two different kinds of naphthalenes: one attached by carbonyl groups and the other attached with aminomethyl groups. The flipping of these two kinds of naphthalenes results in five different conformations, as shown in Fig. 2b. These five conformers have different symmetries and thus different peak patterns in the [1]H NMR spectra: conformers I and V have C$_2$ symmetry and thus should give rise to four doublets for their aromatic signals; conformers II and III share C$_1$ symmetry and 16 doublets in the aromatic regions; conformer IV possesses C$_i$ symmetry and should give rise to eight aromatic signals. Provided that the interconversion of these conformers is slow on the NMR timescale, it is possible to directly differentiate at least some of them in the [1]H NMR spectrum of macrocycle **1**.

As shown in Fig. 2d, eight doublets are observed in the aromatic region of the ¹H NMR spectrum of **1** in CD₂Cl₂. Careful inspection of this spectrum indicates that these doublets should belong to two different species because their integrals have different ratios. That is, each species has four doublets for their aromatic protons (Supplementary Fig. 1). This is only possible when conformers I (C₂ symmetry) and V (C₂ symmetry) coexist in solution. In addition, this result also indicates that the interconversion between these two conformers is slow on the NMR timescale at room temperature, which is consistent with the split signals of the bridging methylene groups (Supplementary Fig. 1). Energy-minimized structures indeed support that among the five conformers, conformers I and V have the most stable conformations (Fig. 2c). Conformers I and V have different structural arrangements and may have different nuclear Overhauser effect (NOE) signals. Therefore, the two structures may be distinguished and assigned by ROESY NMR experiments. Surprisingly, similar NOE peaks were observed (Supplementary Figs. 2 and 3). Nevertheless, the two conformers can be unambiguously assigned according to the X-ray single crystal structure of the host-guest complex and their NMR spectra (see below). In CD₂Cl₂, the minor species is assigned to conformer I, and the major species is assigned to conformer V. Conformer I and conformer V also coexist in other deuterated solvents, but their ratios are slightly altered in different solvents ([**I**]/[**V**] is 0.70, 0.90, 0.97 and 1.47 in CD₂Cl₂, CDCl₃, Cl₂CDCDCl₂ and Tetrahydrofuran-$d_8$ (THF-$d_8$), respectively; Fig. 2d and Supplementary Fig. 4).

Conformations of oxatub[4]arene even undergo quick interconversion at very low temperature[24], but the conformation exchange of **1** is slow at room temperature, which is likely caused by the amide groups. The C–N bond in the amide moiety possesses partial double bond character and is known to have a high rotational barrier[33]. Conformational exchange in **1** involves flipping the naphthalene and may also involve rotation around the C–N bonds, resulting in a high rotational barrier. Variable-temperature ¹H NMR experiments in Cl₂CDCDCl₂ (Fig. 2e and Supplementary Fig. 5) were performed to determine the barrier of conformational interconversion. These aromatic signals gradually broaden as the temperature increases, and coalescence was reached at approximately 100 °C. Further increasing the temperature led to the observation of a single set of signals. The activation free energy ($\Delta G^{\ddagger}$) was estimated to be 74 kJ/mol by using the coalescence temperature ($T_c$) and the chemical shifts ($\delta v$) at 30 °C and by using the standard equation[34] ($\Delta G^{\ddagger} = 8.314 T_c[22.96 + \log(T_c/\delta v)]$; for details, see Supplementary Fig. 5). This conformational exchange barrier is rather close to the C–N bond rotational barrier of DMF (71 kJ/mol)[33], supporting the above discussion. In addition, the equilibrium constants between the two conformers at different temperatures were also measured (for the equilibrium between conformer V and conformer I, $K_{eq} = 0.97, 0.99, 1.01$ for 30, 50, and 70 °C, respectively). According to the van't Hoff equation (Supplementary Fig. 6), the difference in enthalpy and entropy between the two conformations can be obtained ($\Delta H = 0.87$ kJ/mol, $-T\Delta S = -0.78$ kJ/mol). These results suggest that conformer I has slightly higher structural freedom but suffers slightly more strain than conformer V.

**Direct observation of conformational selection.** In contrast to the structure of oxatub[4]arene, the cavity of **1** contains hydrogen bonding donors. Energy-minimized structures (Fig. 2c) show amide protons pointing towards the cavity of conformers I and V defined by four naphthalenes. This is rather similar to the amide naphthotubes we reported earlier[28], but more hydrogen bonding

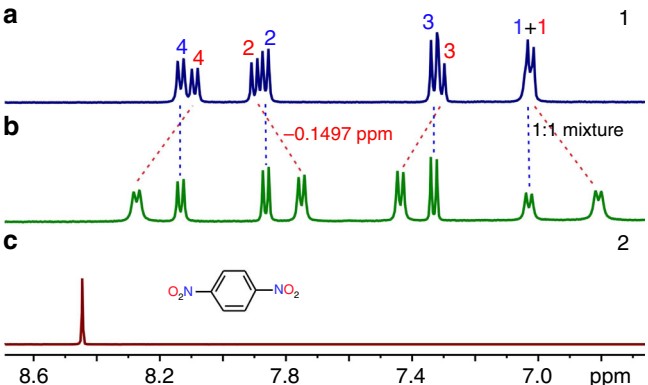

**Fig. 3 Host-guest binding between macrocycle 1 and guest 2.** Partial ¹H NMR spectra (500 MHz, CD₂Cl₂, 2.0 mM, 25 °C) of (**a**) **1**, (**c**) **2** and (**b**) their 1:1 mixture. The colors of the numberings on the NMR peaks are the same as those on the structures of the conformations (Fig. 2b).

donors (N–H protons) are located inside the cavity of **1**. A good guest should satisfy all the hydrogen bonding sites. For conformers I and V, the cavity size and the arrangement of the four amide protons are different. This may cause the two conformers to have drastically different binding affinities to the same guest. Ideally for the conformational selection mechanism, a guest will only bind to one of the conformations and will not bind to the other at all.

1,4-Dinitrobenzene (**2**, Fig. 2a) was found to be such a guest. The addition of **2** into the solution of **1** resulted in an obvious upfield shift of the signals of conformer I in the NMR spectra (Fig. 3 and Supplementary Figs. 7–9). In contrast, the signals of conformer V undergo a rather minor change, which is similar to the case of an acyclic amide compound (Supplementary Fig. 10). This suggests that there is no specific binding between guest **2** and conformer V. Furthermore, no signals for free guest and free conformer I are detected, suggesting the guest exchange kinetics of complex **2@1-I** is fast on the NMR timescale even at -30 °C (Supplementary Figs. 7 and 11). This is in contrast to the slow interconversion kinetics between the two conformers. Nevertheless, the signals of guest **2** are not detected, which should be broadened and disappear into the baseline (Supplementary Fig. 7). With increasing concentration of **2**, conformer V is slowly converted to conformer I (Supplementary Fig. 7). In addition, the amide protons of conformer I shift downfield. This information suggests that guest **2** should be encapsulated inside the cavity of conformer I.

A single crystal of **2@1**, suitable for X-ray crystallography, was obtained by vapor diffusion of pentane into a CH₂Cl₂ solution of macrocycle **1** and guest **2** (1:1 stoichiometry). As shown in Fig. 4a, guest **2** fits snuggly into the cavity of conformer I: the two nitro groups of the guest are well accommodated by the amide protons of the host through hydrogen bonding; the aromatic protons of the guest form C-H···π interactions with the electron-rich naphthalenes (the ones attached by aminomethyl groups rather than by carbonyl groups) of the host; in addition, all four hydrogen bonds involved between the host and the guest have similar angles and distances, indicating that they are in perfect cooperation and balance. The host conformer selected by guest **2** in solution should also be conformer I. This assignment is further supported by the DFT calculations (Fig. 4b). Complex **2@1-I** is more stable than complex **2@1-V** by 31 kJ/mol. The calculated structure of **2@1-I** is very similar to its crystal structure. In contrast, only one short hydrogen bond is detected in complex **2@1-V**, and the host structure is twisted. That is, the cavity of conformer V does not provide a good environment for the

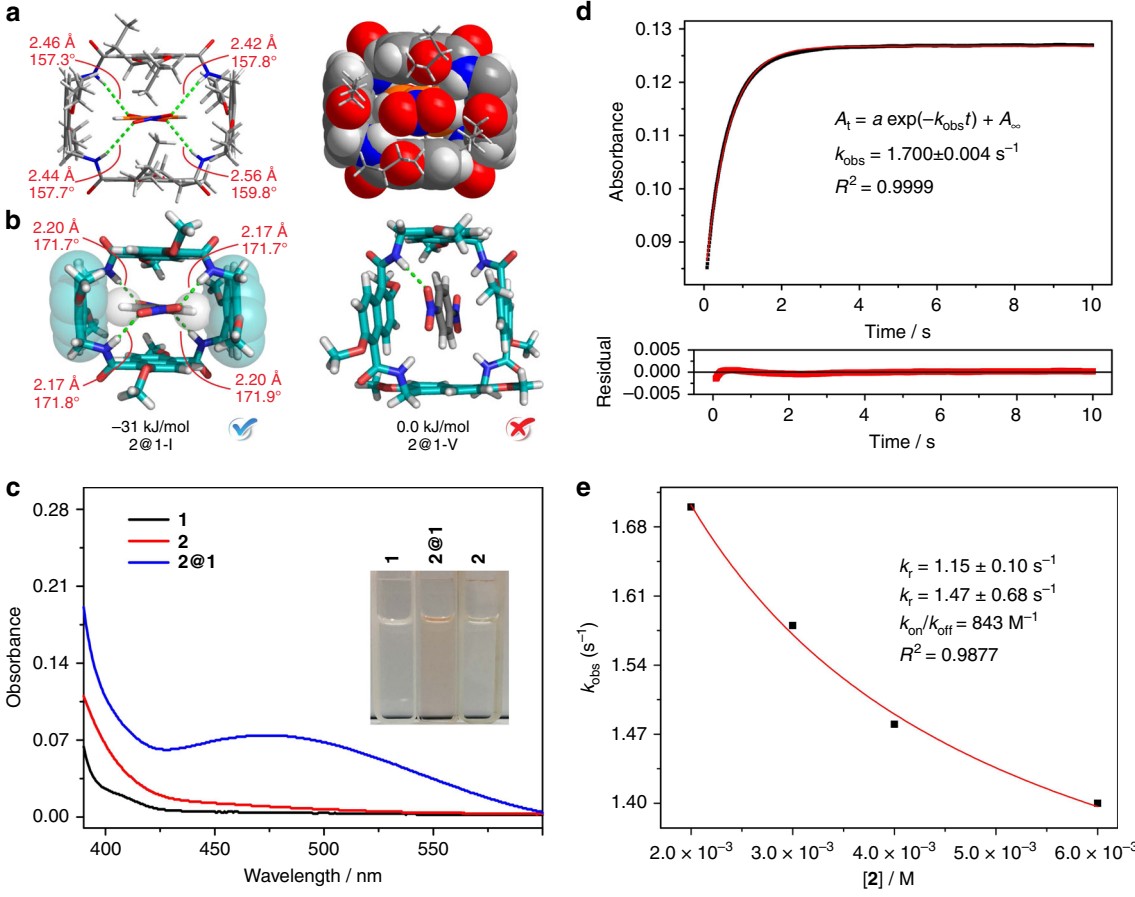

**Fig. 4 Binding mode and binding kinetics between macrocycle 1 and guest 2. a** Single-crystal structure of **2@1-I**. **b** Energy-minimized structures of **2@1-I** and **2@1-V** calculated by DFT at the wB97XD/6-31 G(d) level of theory in CH$_2$Cl$_2$. The butyl groups are shortened to methyl groups for convenience. **c** UV-vis absorption spectra (1.0 mM, CH$_2$Cl$_2$, 25 °C) of **1**, **2@1** and **2**. Inset: photos of the three solutions. **d** Evolution of the UV-vis absorbance intensity at 480 nm with time monitored by a stopped-flow spectrometer after mixing **1** (2.0 mM, CH$_2$Cl$_2$) and **2** (4.0 mM, CH$_2$Cl$_2$). The red solid line represents the fitted curve for a single-exponential function (where $A_t$ is the absorbance intensity (480 nm) at any time, $A_\infty$ is the final value of absorbance intensity (480 nm), and $k_{obs}$ is the observed first-order rate constant). Residual for the fitting is shown in the bottom panel. **e** Nonlinear fitting of $k_{obs}$ with [**2**] according to Eq. (1).

accommodation of **2**. This explains why conformer V does not bind **2** at all.

The above results provide the following information: (a) conformers I and V coexist in solution, with both at detectable amounts; (b) the conformational interconversion between conformers I and V is slower than guest exchange in complex **2@1-I**; (c) guest **2** binds only to conformer I and not to conformer V. Therefore, the host-guest pair between **1** and **2** clearly follows a conformational selection binding mechanism. In addition, one may note that the cavity of the energy-minimized structure of conformer I of free **1** is slightly different from that of **2@1-I**. That is, once conformer I is selected by guest **2**, local induced-fit conformational change is also necessary to adjust the host to be perfectly complementary to the guest. Therefore, a conformational selection mechanism followed by a local induced fit may be more common for interpreting conformational changes in molecular recognition[35].

**Kinetic aspect of conformational selection.** The host-guest pair between **1** and **2** clearly follows the binding mechanism of conformational selection and thus can be used as a simplified model system to study the kinetic and thermodynamic aspects of conformational selection in detail. The binding kinetics between **1** and **2** are rather fast and cannot be monitored by $^1$H NMR

spectroscopy. However, **1** and **2** form a charge-transfer complex, which exhibits a charge-transfer peak centered at approximately 480 nm in the UV-vis spectra (Fig. 4c). Therefore, the binding kinetics were followed by monitoring the charge-transfer absorption at 480 nm with a stopped-flow spectrometer after mixing **1** (2.0 mM) with different concentrations of **2**. The binding details are ignored, and the observed rate constants ($k_{obs}$) were obtained. All the trace curves are fitted very well according to a single exponential equation (Fig. 4d and Supplementary Figs. 15–17)[36].

The observed rate constants are shown in Table 1. Obviously, the observed rate constants decrease with increasing concentrations of **2**. This is a typical kinetic characteristic for the mechanism of conformational selection. Moreover, these data can be well fitted according to Eq. (1) (Fig. 4e). Equation (1) is

**Table 1 The observed rate constants ($k_{obs}$, s$^{-1}$) with different concentrations of 2.**

| [**2**]/(M) | $2.0 \times 10^{-3}$ | $3.0 \times 10^{-3}$ | $4.0 \times 10^{-3}$ | $6.0 \times 10^{-3}$ |
|---|---|---|---|---|
| $k_{obs}$/s$^{-1}$ | 1.700 ± 0.004 | 1.578 ± 0.002 | 1.484 ± 0.001 | 1.399 ± 0.001 |

often used qualitatively to determine whether the ligand binding processes of bioreceptors follow a conformational selection mechanism. In the present case, Eq. (1) is even quantitatively obeyed, as the binding pair between **1** and **2** strictly follows the binding mechanism of conformational selection.

With nonlinear fitting (Fig. 4e), the conformational inter-conversion rate constants can be obtained: the rate constant ($k_r$) from conformer V to conformer I is $1.15\,\mathrm{s}^{-1}$, while the backward rate constant ($k_{-r}$) is $1.47\,\mathrm{s}^{-1}$. Therefore, the equilibrium constant between conformer V and I was calculated to be $K = k_r / k_{-r} = 0.78$. This is similar to the equilibrium constant (0.70) obtained by using the integrals of the two conformers in $CD_2Cl_2$ (Supplementary Fig. 4). In addition, the nonlinear fitting also gave rise to $k_{on}/k_{off}$ ($843\,\mathrm{M}^{-1}$), which is the association constant between conformer I and **2**. This number is on the same order of magnitude as the association constant obtained with thermodynamic fitting (see below).

All these results further corroborate that the binding pair between **1** and **2** strictly follows the conformational selection mechanism. Not only can the kinetic data be well fitted with Eq. (1), but the thermodynamic data can also be calculated, and these data are fully consistent with those obtained with the thermodynamic experiments. This indicates that Eq. (1) accurately describes the kinetic and thermodynamic aspects of conformational selection.

**Thermodynamic aspect of conformational selection.** Proteins often exist as a conformational ensemble. Even though a conformational selection binding mechanism is followed, the same guest may bind two or more conformations simultaneously but show a preference to one over the others. Due to the complexity of the biological systems and limited analytical tools to reveal the binding details, the binding affinity of a ligand to the inactive conformation is often not studied. Thus, the thermodynamic aspect of conformational selection, such as the selectivity between the active and inactive conformations, is largely ignored. With the current simplified supramolecular system in hands, the thermodynamic aspect of conformational selection can be studied as well.

As shown in Fig. 5, three equilibria are involved if a guest can bind both conformers I and V. The association constants of the

With this factor, the thermodynamic fidelity of conformational selection can be quantified. When $\alpha$ is large, one conformation is thermodynamically more preferred over the other. However, how are the association constants of the same guest binding to the two interconvertible conformations simultaneously determined? An HPLC method has been developed for a dynamic combinatorial library[37], but this method is not suitable for conformational equilibrium. Here, new equations are then developed to describe this process.

To simultaneously determine the association constants of the same guest to the two conformations, two mathematical models were developed by using the $^1$H NMR signals and integrals of the two conformations ($[\mathbf{H}]_t$ and $[\mathbf{G}]_t$ are the total concentration of the host and the guest, respectively; $\delta_{obs}$ is the chemical shifts of the proton of interest; $\delta_F$ and $\delta_B$ are the chemical shifts of the proton of interest in their free and bound states; $[\mathbf{I}]_t$ and $[\mathbf{V}]_t$ are the total molar concentration of each conformer). In the first model, the chemical shifts ($\delta_{obs}$)[38] of the two host conformations can be expressed as a function of the total concentration of the added guest ($[\mathbf{G}]_t$). Thus, Eqs. (6) and (7) were developed for conformer I and conformer V, respectively. The derivation details are included in Supplementary Note 1. The two equations can be fitted by using the global fitting method with sharing their parameters to give the two association constants ($K_1$, $K_2$) simultaneously[39]. For the second mathematical model, the ratio $[\mathbf{I}]_t/[\mathbf{V}]_t$ of the total concentrations of the two conformers is expressed as a function of $[\mathbf{G}]_t$, as shown in Eq. (8). The ratio $[\mathbf{I}]_t/[\mathbf{V}]_t$ can be easily obtained from their $^1$H NMR integrals because guest exchange is faster than conformational interconversion and the latter is slow on the NMR timescale. Again, nonlinear fitting of the data according to Eq. (8) can afford the two association constants simultaneously. These two mathematical models should both work, and their results can be used to corroborate each other.

$$K_1 = \frac{[\mathbf{G@I}]}{[\mathbf{I}][\mathbf{G}]} \tag{3}$$

$$K_2 = \frac{[\mathbf{G@V}]}{[\mathbf{V}][\mathbf{G}]} \tag{4}$$

$$K = \frac{[\mathbf{V}]}{[\mathbf{I}]} \tag{5}$$

$$\delta_{obs} = \frac{2(K_1 + KK_2)\delta_F + \delta_B K_1 \left\{ -\left(1 + K + (K_1 + KK_2)([\mathbf{H}]_t - [\mathbf{G}]_t)\right) + \sqrt{\left(1 + K + (K_1 + KK_2)([\mathbf{H}]_t - [\mathbf{G}]_t)\right)^2 + 4(K_1 + KK_2)(1 + K)[\mathbf{G}]_t} \right\}}{2(K_1 + KK_2) + K_1 \left\{ -\left(1 + K + (K_1 + KK_2)([\mathbf{H}]_t - [\mathbf{G}]_t)\right) + \sqrt{\left(1 + K + (K_1 + KK_2)([\mathbf{H}]_t - [\mathbf{G}]_t)\right)^2 + 4(K_1 + KK_2)(1 + K)[\mathbf{G}]_t} \right\}} \tag{6}$$

$$\delta_{obs} = \frac{2(K_1/K + K_2)\delta_F + \delta_B K_2 \left\{ -\left(1 + 1/K + (K_1/K + K_2)([\mathbf{H}]_t - [\mathbf{G}]_t)\right) + \sqrt{\left(1 + 1/K + (K_1/K + K_2)([\mathbf{H}]_t - [\mathbf{G}]_t)\right)^2 + 4(K/K + K_2)(1 + 1/K)[\mathbf{G}]_t} \right\}}{2(K_1/K + K_2) + K_2 \left\{ -\left(1 + 1/K + (K_1/K + K_2)([\mathbf{H}]_t - [\mathbf{G}]_t)\right) + \sqrt{\left(1 + 1/K + (K_1/K + K_2)([\mathbf{H}]_t - [\mathbf{G}]_t)\right)^2 + 4(K_1/K + K_2)(1 + 1/K)[\mathbf{G}]_t} \right\}} \tag{7}$$

$$\frac{[\mathbf{I}]_t}{[\mathbf{V}]_t} = \frac{2(K_1 + KK_2) + K_1 \left\{ -\left(1 + K + (K_1 + KK_2)([\mathbf{H}]_t - [\mathbf{G}]_t)\right) + \sqrt{\left(1 + K + (K_1 + KK_2)([\mathbf{H}]_t - [\mathbf{G}]_t)\right)^2 + 4(K_1 + KK_2)(1 + K)[\mathbf{G}]_t} \right\}}{2K(K_1 + KK_2) + KK_2 \left\{ -\left(1 + K + (K_1 + KK_2)([\mathbf{H}]_t - [\mathbf{G}]_t)\right) + \sqrt{\left(1 + K + (K_1 + KK_2)([\mathbf{H}]_t - [\mathbf{G}]_t)\right)^2 + 4(K_1 + KK_2)(1 + K)[\mathbf{G}]_t} \right\}} \tag{8}$$

same guest to the two conformers are $K_1$ and $K_2$, which can be defined in Eqs. (3) and (4), respectively. The equilibrium constant ($K$) between the two conformations is defined in Eq. (5). The molar concentration of **G@I** and **G@V** are $[\mathbf{G@I}]$ and $[\mathbf{G@V}]$, respectively. A conformational selectivity factor ($\alpha$) may be defined as $\alpha = K_1/K_2$.

In addition to guest **2**, dimethyl terephthalate (**3**), methyl 4-nitrobenzoate (**4**), 4-nitroacetophenone (**5**), and 1,4-diacetylbenzene (**6**) are guests for macrocycle **1** as well (Supplementary Figs. 18–21). However, nitrobenzene (**7**) is not bound by either conformer, although it is a substructure of **2** (Supplementary

Fig. 22). Different from **2** (Figs. 3 and 6a), guests **3–6** can be complexed by both conformers I and V, as indicated by the obvious shift in their ¹H NMR signals with the addition of guests (Fig. 6b, c and Supplementary Figures 18–21). The signals of conformer I generally undergo a larger shift, suggesting that conformer I is a better binder than conformer V for all these guests. This is supported by the changes in the ratio of two conformers (Supplementary Fig. 23). There is a lack of macroscopic signals for guests **3–6**, such as the charge-transfer signals observed for **1** and **2**, to confirm the binding mechanism kinetically. However, a conformational selection mechanism should be followed for these guests as well: the conformational exchange of **1** requires flipping of naphthalene, which has to pass through the cavity; once the cavity of one conformer is occupied by a guest, it cannot be converted to the other conformer; as a consequence, conformational interconversion must occur before guest binding, and the induced fit mechanism is not possible.

The binding of guests **3–6** to macrocycle **1** obeys the two mathematical models discussed above. Therefore, the ¹H NMR data obtained by titrating guests **3–6** into a solution of **1** were fitted (Fig. 6d, e and Supplementary Figs. 24–35) according to the two mathematical models (Eqs. (6), (7) and (8)). The association constants to the two conformers are listed in Table 2.

Generally, the two models give rather similar association constants, indicating that they are both reliable. The association

$$G@I \xrightleftharpoons[]{K_1} G + I \xrightleftharpoons[]{K} V + G \xrightleftharpoons[]{K_2} G@V$$

**Fig. 5 Conformational interconversion and complexation equilibria.** $K_1$ and $K_2$ are the association constants of a guest with conformers I and V, respectively; $K$ is the equilibrium constant between the two conformations.

constants to conformer I are highly dependent on the functional groups, showing the following trend: nitro > ester > ketone. That is, guest **4** with one nitro group and one ester group shows the highest binding affinity to conformer I. However, the association constants to conformer V follow a different trend: ester > ketone > nitro. Guest **3** with two ester groups shows the highest binding affinity to conformer V. This leads to different selectivity of these four guests to conformer I over conformer V. The conformational selectivity factors were calculated according to the above definition. As shown in Table 2, guest **4** shows the highest conformational selectivity factor among guests **3–6**. Guest **4** has one nitro group and one ester group and can participate in four hydrogen bonds with the four amide N-H protons of conformer I. The number and strength of the hydrogen bonds is likely the underlying reason for the binding selectivity between the conformers. The hydrogen-bonding donating and electron-withdrawing ability of the three functional groups of the guests follow this order: nitro > ester > ketone. This results in the following order of conformational selectivity for guests **3–6**: **4** > **5** > **3** > **6**.

Equations (6)–(8) can also be applied to the binding pair between **1** and **2**. With Eqs. (6) and (7), the association constants ($K_1$ and $K_2$) for conformers I and V were determined to be 1473 and 9 M⁻¹ (Supplementary Figs. 36, 37), respectively. This association constant for conformer I is the highest among all the guests, but the association constant for conformer V is the smallest. Consequently, the conformational selectivity factor (164) is the highest. This again supports our analysis of the role of hydrogen bonding in determining the conformational selectivity. $K_2$ is relatively small. By ignoring the binding of guest **2** to conformer V as suggested by the ¹H NMR experiments (Fig. 4 and Supplementary Fig. 7), Eq. (6) can be adjusted and

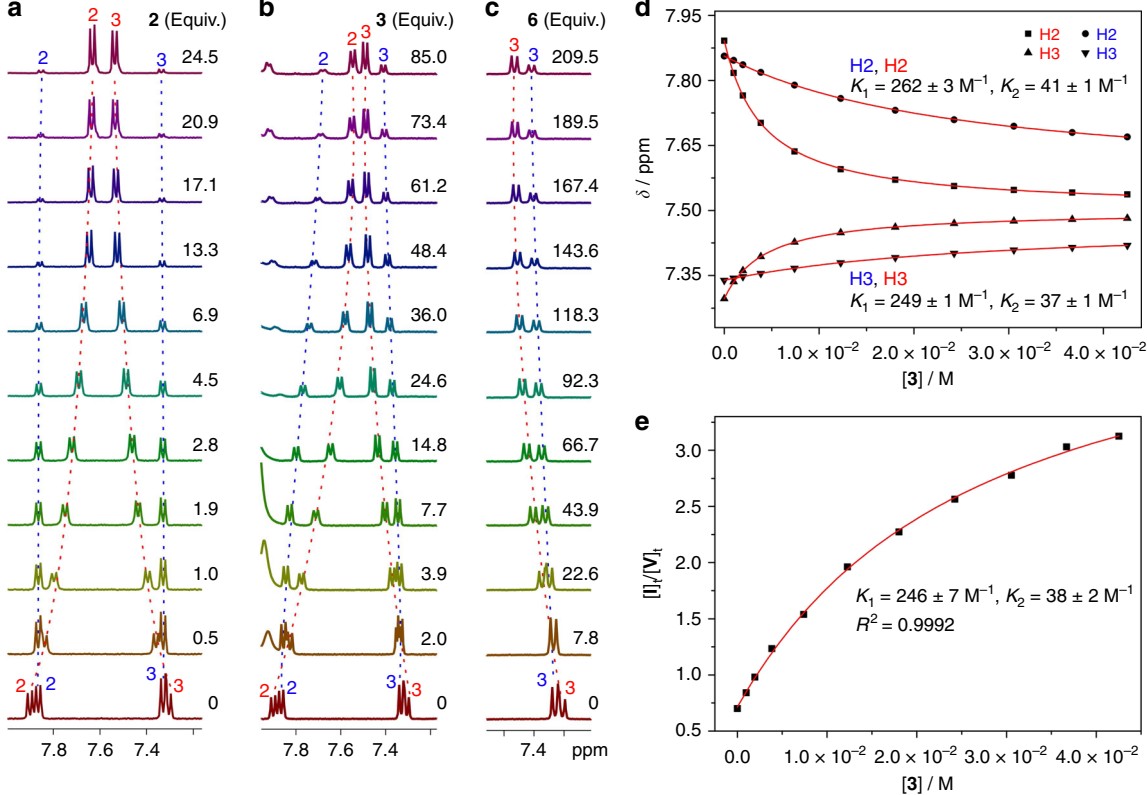

**Fig. 6 NMR titration and nonlinear fitting data. a–c** Partial ¹H NMR spectra (500 MHz, 25 °C) of **1** (0.5 mM) during titration with guests **2**, **3** and **6**, respectively, in CD₂Cl₂. **d, e** Nonlinear curve-fittings of the ¹H NMR data of **1** titrated by **3** according to Eqs. (6) and (7) and according to Eq. (8), respectively.

**Table 2 Association constant $K_a$ values ($M^{-1}$) of macrocycle 1 with neutral molecules at 25 °C as determined by NMR titration ($CD_2Cl_2$) according to Eqs. (6)–(8).**

| Guests | $K_1{}^a$ | $K_2{}^a$ | $\alpha$ | $K_1{}^b$ | $K_2{}^b$ | $\alpha$ |
|---|---|---|---|---|---|---|
| 2 | 1473 ± 17 | 9 ± 1 | 164 | 1241 ± 47 | –c | — |
| 3 | 258 ± 2 | 40 ± 1 | 6.5 | 238 ± 8 | 36 ± 2 | 6.6 |
| 4 | 845 ± 9 | 20 ± 1 | 42 | 732 ± 1 | 9 ± 1 | 81 |
| 5 | 193 ± 17 | 12 ± 2 | 16 | 162 ± 13 | 7 ± 1 | 23 |
| 6 | 43 ± 5 | 16 ± 5 | 2.7 | 31 ± 7 | 8 ± 3 | 3.8 |

aNon-linear curve-fitting of the NMR data according to Eqs. (6) and (7).
bNon-linear curve-fitting of the NMR data according to Eq. (8).
cThe obtained $K_2$ is not reasonable because it is negative (−6), presumably because the association constant of conformer V is too low to be accurately determined.

used alone for the nonlinear fitting of the data (Supplementary Fig. 38). The obtained $K_1$ value is 1467 $M^{-1}$, which is very similar to that reported above. This is also consistent with the association constant obtained by the integration method ($K_1 = 1206\ M^{-1}$, Supplementary Fig. 39). These results indicate that the association constant of **2** to conformer V is quite small. With equation (8), $K_1$ was determined to be 1241 $M^{-1}$ (Supplementary Figs. 40 and 41), which is quite close to that determined from Eqs. (6) and (7). However, the obtained $K_2$ is not reasonable because it is a negative number. This further supports that the binding affinity to conformer V is quite weak. This is in line with the crystal and energy-minimized structures of the complexes involving guest **2** (Fig. 4a, b). The binding pair between host **1** and guest **2** is the extreme case for the conformational selection mechanism, but its thermodynamic data can still be fitted according to Eqs. (6)–(8). Consequently, Eqs. (6)–(8) are good mathematical models to study the thermodynamic aspects of the binding pairs that follow a conformational selection mechanism.

## Discussion

In summary, we report a simple macrocyclic receptor that strictly follows the binding mechanism of conformational selection. The receptor possesses the structural feature of both oxatub[4]arene and amide naphthotube. Naphthalene flipping results in five possible conformations with drastically different cavities; four amide N-H protons are directed into the cavity, realizing high guest-binding selectivity. Two of the five conformations are found to coexist in solution in the absence of a guest. These two conformations undergo rather slow interconversion with a barrier of 74 kJ/mol. 1,4-Dinitrobenzene is found to be able to predominantly bind one conformer over the other, with fast guest exchange kinetics on the NMR timescale. Thus, the binding mechanism can be unambiguously assigned to conformational selection through thermodynamic NMR experiments, which is further supported by kinetic experiments. The kinetic equation for conformational selection is quantitatively obeyed by the present system. In addition, several similar guests are found to bind both conformers and should also follow a conformational selection mechanism. Two mathematical models are developed to simultaneously determine the association constants of the same guest to two conformers in a complex equilibrium system. The conformational selectivity factor is then defined and calculated to quantify the thermodynamic fidelity of conformational selection.

Conformational change in molecular recognition is the central topic in biophysics. Debates on the two limiting mechanisms, conformational selection and induced fit, have endured for over 5 decades, with induced fit being favored for very long time[4]. Only recently has conformational selection started to gain more

attention and been found to be essential in interpreting the complex biological phenomena. In contrast, the discussion of conformational selection is still rare in supramolecular community[18–20]. When supramolecular systems become more and more complex, a conformational selection mechanism may be involved as well. The current research provides an in-depth analysis on the kinetic and thermodynamic aspects of conformational selection, and should be helpful in understanding complex supramolecular systems with a conformational selection mechanism.

## Methods

**General**. All the reagents involved in this research were commercially available and used without further purification unless otherwise noted. $^1H$, $^{13}C$ NMR, $^1H$–$^1H$ COSY and $^1H$-$^1H$ ROESY NMR spectra were recorded on Bruker Avance-400 (500, 600) spectrometers. Electrospray-ionization time-of-flight high-resolution mass spectrometry (ESI-TOF-HRMS) experiments were conducted on an applied Q EXACTIVE mass spectrometry system. Absorption spectra were recorded on a Hitachi U-2600 UV–vis spectrophotometer.

**Synthesis and characterization**. Synthesis and the corresponding characterization data are provided in the Supplementary Methods.

**Stopped-flow experiments**. Stopped-flow experiments were performed with a SX20 system (Applied Photophysics). The samples were kept at 25.0 °C for at least 5 min and were then mixed in a 1:1 ratio. For each experiment, at least 5 individual kinetic traces were averaged. Each averaged of the kinetic data were then fitted to a single-exponential function. The final concentrations of stopped-flow experiments are half of the concentrations in each syringe.

**Determination of the association constants by NMR titrations**. To determine the association constants, NMR titrations were performed at 298 K by titrating the guests to the solution of the host in $CD_2Cl_2$ with a fixed host concentration (0.5 mM). Through sharing parameters, global non-linear curve-fitting was performed on the plots of $\delta_{obs}$ of conformers I and V as a function of $[G]_t$ to give the association constants ($K_1$ and $K_2$). Non-linear curve-fitting was also performed on the plot of $[I]_t/[V]_t$ (obtained from $^1H$ NMR integrals of the two conformers) as a function of $[G]_t$ to give the association constants ($K_1$ and $K_2$). NMR titration and nonlinear fitting data are shown in Fig. 6 and Supplementary Figs. 24–41.

## Data availability

The X-ray crystallographic coordinates for structures reported in this study have been deposited at the Cambridge Crystallographic Data Centre (CCDC), under deposition numbers 1950443 (2@1-I). These data can be obtained free of charge from the Cambridge Crystallographic Data Centre via www.ccdc.cam.ac.uk/data_request/cif. All other data supporting the findings of this study are available within the Article and its Supplementary Information and/or from the corresponding author upon reasonable request.

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

## Acknowledgements

This research was financially supported by the National Natural Science Foundation of China (Nos. 21772083, 21822104), the Shenzhen Science and Technology Innovation Committee (KQJSCX20170728162528382, JCYJ20180504165810828), the Shenzhen Nobel Prize Scientists Laboratory Project (C17213101), Guangdong Provincial Key Laboratory of Catalysis (No. 2020B121201002), and the University of Jyväskylä. W.J. acknowledges Shenzhen Education Bureau for the support of "Pengcheng Scholar". The DFT calculations were supported by the Center for Computational Science and Engineering of SUSTech. We are grateful to the technical support from SUSTech-CRF.

## Author contributions

W.J. and L.-P.Y. conceived and designed the experiments. L.-P.Y. carried out most of the experimental work. L.Z. contributed to host synthesis. M.Q. and Y.-L.M. performed the DFT calculations. J.S.W. and K.R. solved the crystal structure. H.Z. contributed to the derivation of fitting equations of association constants. W.J. and L.-P.Y. analyzed the data and wrote the manuscript, and all authors commented on it.

## Competing interests

The authors declare no competing interests.
