## [Peer Review File · Nature Communications]

Reviewers' comments:

Reviewer #1 (Remarks to the Author):

The manuscript "A Supramolecular Model System That Strictly Follows the Binding Mechanism of Conformational Selection" by Jiang and co-workers describes a novel supramolecular model system that allows for distinguishing between the two main biological binding models: the induced fit model and the conformational selection model. In biological systems, these are usually difficult to distinguish as their kinetics can be similar and the thermodynamics are often impossible to determine. In this study, Jiang and co-workers developed a supramolecular macrocycle, in which the guest binding strictly follows the conformational selection model. This model system did not only allow them to study the kinetics of conformational selection in detail and validate the existing kinetic models, but also develop a new thermodynamic model to describe the observed equilibria and the preference of one host conformer to the guest by a "conformational selectivity factor".

The study of the system is thorough and the conclusions backed by the necessary analytical data. Stopped-flow experiments were repeated several times and the data of at least five curves fitted simultaneously. In case of titrations, two data sets were fitted simultaneously. Overall this study is scientifically sound, although the wording could be polished in some places to ease the understanding of the complex material. The manuscript references previous literature appropriately.

This study furthermore shows how powerful supramolecular model systems can be to better understand phenomena we observe in biology. Hence, this referee believes this manuscript to be well suited for the broad readership of Nature Communications, because it should not only be of interest to the supramolecular and physical organic chemistry communities, but also biological chemistry and biophysics.

In my opinion, this is a great piece of research and should be published after a few minor revisions:

1. None of the rate or equilibrium constants in the manuscript do have any errors. They can be found in the SI, but should certainly be included in the main text as well. Any data obtained by experimental methods naturally has an error value and the reader shouldn't have to hunt for it. Most errors seem to be reasonably low, so there is no reason to hide them. Errors should especially be included in Figures 4 and 5 and Table 1.

2. The equilibrium arrows for host-guest complex formation in eq. 3 should point into the opposite direction to properly reflect eqs. 4 and 5:

(equilibrium arrows do not type in a text editor. These arrows reflect the correct direction of the top arrow)

Same is true for the equilibrium in the SI, page S30.

3. The derivation of the thermodynamic model in the SI, page S30 onwards, might not be trivial to the broad readership of Nature Communications. Hence, this referee strongly suggests the authors to include more steps, especially between eqs. 8 and 10, eqs. 11 and 12, eqs. 14 and 15 as well as between eqs. 17 and 18.

4. Furthermore, there is a typo in line 297, page S31:

"...the total concentration of the host [H]_t and the host [G]_t..."
should be:

...the total concentration of the host [H]_t and the GUEST [G]_t...

5. Eqs. 1 and 2 in the main text seem to be derived from Reference 14, yet this is not clarified in the main text as it should be.

Reviewer #2 (Remarks to the Author):

In the manuscript the authors present a study aimed at understanding the binding mechanism between a macrocyclic host, structured with oxatub[4]arene modules, and dinitrobenzene. The host has been found in two conformations, characterized structurally through gas-phase DFT calculations and NMR. Kinetics data suggests that the binding mechanism follows a conformational selection profile. I found the study interesting, but rather limited in its scope. More specifically, I do not see how the system can be considered a "biomimetic", thus how the conclusions can be transferable beyond this specific study. Indeed, I do not see how the host's narrow cavity, layered by naphthalene moieties, can be representative (or resembling) a protein binding site, despite the peptide bond-like linkers between the aromatic units.

Furthermore, provided my interpretation of the study is correct, the fact that the ligand binds only one of the two conformers, i.e. the one with an actually open cavity vs. its twisted counterpart, is not entirely surprising as it looks that the dinitrobenzene may only "fit" into the open conformers, therefore excluding the twisted one from the equation. The mechanistic details on how the host cavity opening proceeds, in my view do strictly depend on a more complex balance between the binding affinity and the free energy of interconversion between the conformers in the host and potential interactions between host and guest(s), which could actually initiate the recognition process and then affect the opening of the cavity.

Based on the reasons above, but mostly because of the non-transferability of the results to biomolecular recognition in general, it is my opinion that this work would fit better in a journal specialised in supramolecular chemistry. I would also personally suggest that the claims about the universality of conformational selection as "the" mechanism for ligand binding should be slightly toned down, as well as the claims about the transferability to such a simplistic model to biological settings (see below).

Abstract

Line 12. In the first sentence of the abstract, the authors state "Conformational selection has recently been considered the most appropriate mechanism for interpreting the conformational changes that occur during biomolecular recognition". I would argue that this is a bit of an exaggeration, as the mechanisms of molecular recognition and ligand binding are inherently system dependent and on its intrinsic flexibilities, i.e. degrees of structural (or conformational) disorder. Indeed, there is good evidence for conformational selection in biomolecular recognition, as there is also of induced-fit and of mechanisms that can be described as intermediate between the two extremes.

Introduction

Line 36. It is true that denatured and disordered proteins do exist in conformational ensembles, but in my opinion, this is hardly an argument in favour of conformational selection. While I would argue also that the reference from Vogt and Di Cera Biochemistry (2013), supporting conformational selection as the dominant mechanism for ligand binding, hardly applies across all different types of ligands (see IDPs for example, or complex carbohydrates) and of binding events.

Reviewer #3 (Remarks to the Author):

The data is technically sound:

The crystal structure quality is average to low due to low-quality diffraction from very small crystal and the presence of significant disorder of host tails and solvent molecules. Despite all that, this structure determination can be considered at the state-of-the-art level and provides solid proofs of guest intercalation into the host molecule and their interaction.

The paper provides strong evidence for its conclusions:

Yes, see above.

The results are novel:

and interesting enough to be published in top journals but perhaps not in the most top ones.

The manuscript is important to scientists in the specific field:

Yes

Point-by-Point Response

Reviewers' comments:

Reviewer #1 (Remarks to the Author):

The manuscript "A Supramolecular Model System That Strictly Follows the Binding Mechanism of Conformational Selection" by Jiang and co-workers describes a novel supramolecular model system that allows for distinguishing between the two main biological binding models: the induced fit model and the conformational selection model. In biological systems, these are usually difficult to distinguish as their kinetics can be similar and the thermodynamics are often impossible to determine. In this study, Jiang and co-workers developed a supramolecular macrocycle, in which the guest binding strictly follows the conformational selection model. This model system did not only allow them to study the kinetics of conformational selection in detail and validate the existing kinetic models, but also develop a new thermodynamic model to describe the observed equilibria and the preference of one host conformer to the guest by a "conformational selectivity factor".

The study of the system is thorough and the conclusions backed by the necessary analytical data. Stopped-flow experiments were repeated several times and the data of at least five curves fitted simultaneously. In case of titrations, two data sets were fitted simultaneously. Overall this study is scientifically sound, although the wording could be polished in some places to ease the understanding of the complex material. The manuscript references previous literature appropriately.

This study furthermore shows how powerful supramolecular model systems can be to better understand phenomena we observe in biology. Hence, this referee believes this manuscript to be well suited for the broad readership of Nature Communications, because it should not only be of interest to the supramolecular and physical organic chemistry communities, but also biological chemistry and biophysics.

In my opinion, this is a great piece of research and should be published after a few minor revisions:

Response: Thank you very much for the kind recommendation and the valuable suggestions! We have changed the manuscript accordingly.

1. None of the rate or equilibrium constants in the manuscript do have any errors. They can be found in the SI, but should certainly be included in the main text as well. Any data obtained by experimental methods naturally has an error value and the reader shouldn't have to hunt for it. Most errors seem to be reasonably low, so there is no reason to hide them. Errors should especially be included in Figures 4 and 5 and Table 1.

Response: This has been changed accordingly. The errors were included in Figs. 4 and 5 and Table 1.

2. The equilibrium arrows for host-guest complex formation in eq. 3 should point into the opposite direction to properly reflect eqs. 4 and 5:

(equilibrium arrows do not type in a text editor. These arrows reflect the correct direction of the top arrow)

Same is true for the equilibrium in the SI, page S30.

Response: This has been changed accordingly. The arrow direction and the equilibrium equations were changed.

3. The derivation of the thermodynamic model in the SI, page S30 onwards, might not be trivial to the broad readership of Nature Communications. Hence, this referee strongly suggests the authors to include more steps, especially between eqs. 8 and 10, eqs. 11 and 12, eqs. 14 and 15 as well as between eqs. 17 and 18.

Response: This has been changed accordingly. More steps on the derivation were added to clarify the detained process in the SI.

4. Furthermore, there is a typo in line 297, page S31:

"...the total concentration of the host [H]_t and the host [G]_t..."

should be:

...the total concentration of the host [H]_t and the GUEST [G]_t...

Response: This has been changed accordingly.

5. Eqs. 1 and 2 in the main text seem to be derived from Reference 14, yet this is not clarified in the main text as it should be.

Response: Thank you for this point! It is true that Equations 1 and 2 are derived from Ref. 14 which is cross-cited here to clarify this point.

Reviewer #2 (Remarks to the Author):

In the manuscript the authors present a study aimed at understanding the binding mechanism between a macrocyclic host, structured with oxatub[4]arene modules, and

dinitrobenzene. The host has been found in two conformations, characterized structurally through gas-phase DFT calculations and NMR. Kinetics data suggests that the binding mechanism follows a conformational selection profile. I found the study interesting, but rather limited in its scope. More specifically, I do not see how the system can be considered a “biomimetic”, thus how the conclusions can be transferable beyond this specific study. Indeed, I do not see how the host’s narrow cavity, layered by naphthalene moieties, can be representative (or resembling) a protein binding site, despite the peptide bond-like linkers between the aromatic units.

Furthermore, provided my interpretation of the study is correct, the fact that the ligand binds only one of the two conformers, i.e. the one with an actually open cavity vs. its twisted counterpart, is not entirely surprising as it looks that the dinitrobenzene may only “fit” into the open conformers, therefore excluding the twisted one from the equation. The mechanistic details on how the host cavity opening proceeds, in my view do strictly depend on a more complex balance between the binding affinity and the free energy of interconversion between the conformers in the host and potential interactions between host and guest(s), which could actually initiate the recognition process and then affect the opening of the cavity.

Based on the reasons above, but mostly because of the non-transferability of the results to biomolecular recognition in general, it is my opinion that this work would fit better in a journal specialised in supramolecular chemistry. I would also personally suggest that the claims about the universality of conformational selection as “the” mechanism for ligand binding should be slightly toned down, as well as the claims about the transferability to such a simplistic model to biological settings (see below).

Response: Thank you for the critical comments! We found your comments are helpful for us to improve this manuscript. In most cases, we agree with you. Therefore, the manuscript has been thoroughly changed according to your suggestions. All the statements regarding the biomimetic nature of this system and its transferability to biological systems are removed.

However, we still believe this system is quite unique and is of great interest to supramolecular chemists and biophysical chemists. In addition, we have developed several new thermodynamic equations which can well explain the kinetic and thermodynamic aspects of conformational selection. These equations and the methods used in this research are valuable and may be used when one encounters similar (either biological or synthetic) systems.

Abstract

Line 12. In the first sentence of the abstract, the authors state “Conformational selection has recently been considered the most appropriate mechanism for interpreting

the conformational changes that occur during biomolecular recognition". I would argue that this is a bit of an exaggeration, as the mechanisms of molecular recognition and ligand binding are inherently system dependent and on its intrinsic flexibilities, i.e. degrees of structural (or conformational) disorder. Indeed, there is good evidence for conformational selection in biomolecular recognition, as there is also of induced-fit and of mechanisms that can be described as intermediate between the two extremes.

Response: Thank you for bringing this into our attention! Such point of view was indeed reported in the biological literature. However, we agree with the referee, and it is not possible to give a clear-cut evidence for the binding mechanisms of all biological systems. Therefore, this sentence was changed accordingly to avoid any misunderstanding.

Introduction

Line 36. It is true that denatured and disordered proteins do exist in conformational ensembles, but in my opinion, this is hardly an argument in favour of conformational selection. While I would argue also that the reference from Vogt and Di Cera Biochemistry (2013), supporting conformational selection as the dominant mechanism for ligand binding, hardly applies across all different types of ligands (see IDPs for example, or complex carbohydrates) and of binding events.

Response: Thank you for bringing this into our attention! These sentences were changed accordingly to avoid any misunderstanding.

Reviewer #3 (Remarks to the Author):

The data is technically sound:

The crystal structure quality is average to low due to low-quality diffraction from very small crystal and the presence of significant disorder of host tails and solvent molecules. Despite all that, this structure determination can be considered at the state-of-the-art level and provides solid proofs of guest intercalation into the host molecule and their interaction.

The paper provides strong evidence for its conclusions:

Yes, see above.

The results are novel:

and interesting enough to be published in top journals but perhaps not in the most top ones.

The manuscript is important to scientists in the specific field:
Yes

Response: Thank you for the scientific comments and the recommendation!